# Determinants of non-adherence to treatment for tuberculosis in high-income and middle-income settings: a systematic review protocol

Fatima B Wurie,[1,2] Vanessa Cooper,[2] Robert Horne,[2] Andrew C Hayward[1,3]

[1]Institute of Health Informatics, University College London, London, UK
[2]Centre for Behavioural Medicine, School of Pharmacy, University College London, London, UK
[3]Institute of Epidemiology and Health Care, University College London, London, UK

**Correspondence to**
Fatima B Wurie;
f.wurie@ucl.ac.uk

## ABSTRACT

**Introduction** Treatment for tuberculosis (TB) is highly effective if taken according to prescribed schedules. However, many people have difficulty adhering to treatment which can lead to poorer clinical outcomes, the development of drug resistance, increased duration of infectivity and consequent onward transmission of infection. A range of approaches are available to support adherence but in order to target these effectively a better understanding of the predictors of poor adherence is needed. This review aims to highlight the personal, sociocultural and structural factors that may lead to poor adherence in high-income and middle-income settings.

**Methods and analysis** Seven electronic databases, Medline, EMBASE, CINAHL, PsycInfo, The Cochrane Library, Scopus and Web of Science, will be searched for relevant articles using a prespecified search strategy. Observational studies will be targeted to explore factors that influence adherence to treatment in individuals diagnosed with TB. Screening title and abstract followed by full-text screening and critical appraisal will be conducted by two researchers. Data will be extracted using the Population, Exposure, Comparator, Outcomes, Study characteristics framework. For cross-study assessment of strength of evidence for particular risk factors affecting adherence we will use the Grading of Recommendations, Assessment, Development and Evaluation tool modified for prognostic studies. A narrative synthesis of the studies will be compiled. A meta-analysis will be considered if there are sufficient numbers of studies that are homogenous in study design, population and outcomes.

**Dissemination** A draft conceptual framework will be identified that (A) identifies key barriers to adherence at each contextual level (eg, personal, sociocultural, health systems) and (B) maps the relationships, pathways and mechanisms of effect between these factors and adherence outcomes for people with TB. The draft conceptual framework will guide targeting of adherence interventions and further research.

**PROSPERO registration number** CRD42017061049.

## Strengths and limitations of this study

► Systematic review and meta-analysis of personal, sociocultural and structural risk factors which predict poor adherence.
► Will offer highest level of quantitative evidence to guide targeting of adherence interventions.
► Wide variety of direct and indirect measures of adherence may hamper collation of outcomes.

low-incidence countries,[1] yet little is known of the personal, social and cultural factors that drive non-adherence across different groups in these settings. Poor adherence is cited as the primary reason for suboptimal clinical benefit[2] and leads to poorer clinical outcomes, the development of drug resistance, increased duration of infectivity and consequent onward transmission of infection. Directly observed therapy, short course, is the international standard for TB control. The standard 'short-course' regimen to treat drug-sensitive TB is 6 months. For those diagnosed with multidrug-resistant TB (MDR-TB) this regimen increases to 9–20 months and adverse effects are more common under these regimens. Despite the plethora of effective TB regimens that exist the treatment completion rates are low and vary across different groups and may hamper international efforts to control TB. The long duration of any current effective treatment regimen for TB can make it difficult for patients to take their drugs as prescribed. For example, in 2014 in England 73.3% of short-course-treated patients completed treatment under these circumstances by 6–8 months and 84.5% within a year.[3] The outcome is worse for MDR or rifampicin-resistant TB cases notified in 2013, where only 57.8% had done so by 24 months. These findings indicate that the treatment and clinical outcomes are considerably poor.

## BACKGROUND
### Rationale

Internationally, adherence to treatment for tuberculosis (TB) is recognised as a key tenet of the TB Elimination Framework within

Some studies have used quantitative methods to evaluate these approaches for TB, many of which have been conducted in resource-poor settings. Other qualitative studies[4] in resource-poor settings have highlighted that poverty and gender discrimination, the social context, health service factors and personal factors interact affecting adherence to treatment.

The NICE Medicines Adherence Guidelines[5] recommended that support should be tailored to meet the needs of the individual by addressing both the perceptual factors (eg, beliefs about the illness and treatment)[6] and the practical factors (eg, capability, resources and opportunity) influencing the motivation and ability to start and continue with treatment.[7] This can be summarised as a Perceptions and Practicalities Approach.[6] The extent to which these factors exist or are important in predicting non-adherence to TB treatment in resource-rich settings is yet to be explored.

## Objectives

This systematic review aims to identify the personal, sociocultural and structural factors associated with poor adherence to treatment for TB in high-income and middle-income settings. A better understanding of these factors will better inform development of interventions to strengthen a patient-centred approach for the delivery of TB programmes and services.

## Review question

What are the determinants of non-adherence to treatment in patients with TB in high-income and middle-income settings?

## METHODS
### Eligibility criteria
#### Study design/characteristics
*Inclusion criteria*

Empirical studies employing prospective, longitudinal, cross-sectional or retrospective designs. Randomised and non-randomised prospective comparative studies of interventions will be included if any predictors were found to have increased adherence and continuation of TB treatment.

The active TB condition has to be defined in the study using a clinical diagnosis. For studies presenting treatment completion rates, a definition for completion will have to be provided.

Studies conducted in high-income (a gross national income (GNI) per capita of $12476 or more) and upper middle-income settings (a GNI per capita between $4036 and $12475 as calculated using the World Bank Atlas method for the current 2017 fiscal year) will be included in the review.

*Exclusion criteria*

Studies conducted in resource-limited settings (GNI per capita of $1025 or less in 2015 as calculated by the World Bank Atlas methods for the current 2017 fiscal year) will be excluded because reviews in this area have focused already on low-income and low to middle-income settings.[4]

There will be no restrictions on age, gender or ethnicity of participants.

### Participants
Individuals clinically diagnosed with active TB.

### Exposure
The primary exposures of interest are the risk factors that may influence adherence. Thus, studies reporting on patient demographics, knowledge and attitudes, characteristics of TB disease, social characteristics of patients, service-related factors and comorbidities will be included in the review.

### Outcomes
Studies will be included in the review if the primary outcome is non-adherence. Non-adherence will be determined by self-reporting through attendance at follow-up appointments, collecting prescriptions from clinics, pill counts and pharmacy reports, electronic devices (Medication Event Monitoring System caps), urine inspection, testing for drug levels and directly observed therapy attendance or video-observed therapy sessions. Studies that report outcomes such as non-completion of treatment and/or lost to follow-up and/or treatment refusal will also be included.

### Information sources
Electronic databases, Medline, EMBASE, CINAHL, PsycInfo, The Cochrane Library, Scopus and Web of Science, will be searched. The reference lists of relevant systematic reviews will be screened to find primary articles.

### Search strategy
We will carry out medical subject heading (MeSH) terms and keyword searches for TB, treatment adherence and compliance. We will seek expert consultation from a librarian on our draft search strategy, which will combine MeSH and free text terms (including term explosion) for TB. No filters for study type will be applied for TB studies. We will remove editorials, news items and letters. Articles published in English will be included. No limits on year of publication will be applied. We have included a draft strategy for Medline in the online supplementary file.

The list of proposed search terms will be reviewed by all authors and any necessary adjustments will be made prior to running the search. We will review the reference lists of eligible articles and relevant reviews to identify additional papers not indexed in the databases searched.

### Data management
Output from the searched databases will be exported into Endnote V.7.1 and duplicate records will be removed electronically. Screening and extraction will occur in

a Microsoft Access database to ensure that all retrieved references are fully tracked.

## Selection process

For the initial screening stage, two authors (FBW and VC) will select articles by screening the title and abstract to assess whether they fulfil the study eligibility criteria. Two researchers (FBW and VC) will conduct abstract selection and critical appraisal of the full-text articles. To reduce the risk of missing potentially relevant studies, researchers will adopt a more lenient approach at the first level of screening. Both researchers will obtain full-text articles for studies that meet the review inclusion criteria. Reasons for rejection of articles during both the initial screening and at the full-text screening process will be noted and any discrepancies will be discussed by FBW and VC and consultation with ACH and RH will be done if necessary.

## Data extraction

We will use the Population, Exposure, Comparator, Outcomes, Study characteristics framework to systematise data extraction. Data will be extracted using a standardised template containing information on each of the following five domains:

1. Population: characteristics of the study population (clinically diagnosed with active TB and recommended for treatment), recruitment and sampling methods, inclusion/exclusion criteria.
2. Exposure: any risk factors that may influence adherence. Includes patient demographics (age and sex distribution, ethnicity), bacille Calmette-Guerin vaccination status, knowledge and attitudes of TB, characteristics of TB (including drug-resistant strain status, coinfection status), social characteristics of patients, service-related factors, interventions and comorbidities, number of exposed subjects, any exclusions.
3. Comparators: identification and definition of unexposed subjects, any exclusions.
4. Outcomes: definition and identification of adherence levels for TB, non-completion of concomitant treatment, loss to follow-up, treatment refusal, number of subjects, any exclusions, length of follow-up.
5. Study characteristics: authors, publication year, setting/source of participants, design, period of study, length of follow-up time (if relevant), aims and objectives. Unadjusted and fully adjusted effect estimates for the association between TB and adherence will be recorded. Details of confounders measured will also be noted. Any results from additional stratified analyses will also be recorded.

We will consider contacting corresponding authors to obtain any missing information using a standardised email template.

## Risk of bias quality assessment (in individual studies)

Risk of bias domains will be used from the Cochrane Collaboration for specific observational study designs.

Assessment of risk of bias of individual studies and outcomes will be conducted by two reviewers independently and will subsequently discuss among all authors for arbitration. For cross-study assessment of strength of evidence for particular risk factors affecting adherence, the Grading of Recommendations, Assessment, Development and Evaluation (GRADE) tool modified for prognostic studies[8] will be used. Specifically, differential outcome measurement in exposed and unexposed cohort populations, incomplete follow-up, failure to control for confounding, difference in measurement of exposure, and selection of exposed and unexposed in cohort studies from different populations will be examined. We will examine each outcome for risks of bias, inconsistency, indirectness, imprecision, publication bias and any additional domains deemed appropriate. We will prioritise direct objective measures of adherence, which are less prone to reporting bias.

## Strategy for data synthesis

A high proportion of studies either reporting adherence during the first 2 months (initiation phase of treatment) or throughout treatment and use of wide variety of instruments are anticipated. Adherence measures at 2 and 6 months time points will be measured separately. Use of different instruments will be reported separately. Any consistency of identified risk factors across instruments and for the different time periods will be reported separately. Further analyses of patients with MDR-TB as well as considering MDR-TB as a risk factor for adherence outcomes will be conducted. Subanalyses to assess whether treatment regimens are predicators of non-adherence will be performed. A narrative synthesis of the studies will be compiled, including a consideration of the socioeconomic context in which included interventions were implemented and other critical factors, such as the drug resistance profiles of the study population. The evidence tables will be arranged and divided according to the different treatment durations and regimens for treatment of TB.

## Meta-analysis

If there are sufficient numbers of studies that are homogenous in study design, population and outcomes we will obtain a pooled effect estimate. The choice of fixed or random effects model will be guided by the level of statistical heterogeneity (assessed using the $I^2$ statistic). A P value for $I^2$ less than 0.05 will indicate that heterogeneity among the group of studies being analysed was significant. If the $I^2$ statistic is greater than 50% (with $P<0.05$) for each treatment outcome, a random effects analysis, incorporating the impact of both chance and heterogeneity among study populations and study design, will be chosen over the fixed effects alternative, which assumes that differences among study outcomes are due entirely to chance. We will use STATA to conduct our meta-analysis.

To assess study quality, we will use the GRADE approach, in which quality of the body of evidence is examined for

each outcome rather than by individual study. We will use GRADEpro software to create tables for summary of findings.

## Ethics and dissemination

Ethical review is not required as this study is a systematic review. It is our intention to submit the findings of this review to a peer-reviewed journal and to present at national and international symposia. Based on the results of the systematic review, we will develop a draft conceptual framework that (A) identifies key barriers to adherence at each contextual level (eg, personal, sociocultural, health systems) and (B) maps the relationships, pathways and mechanisms of effect between these factors and adherence outcomes for patients with TB on treatment. The draft conceptual framework will guide research questions and formative primary research to understand the factors that influence irregular patterns and non-adherence.

This protocol has been prepared using the Preferred Reporting Items for Systematic Review and Meta-Analysis Protocols guidelines.[9]

**Contributors** ACH and FBW conceived the idea, and planned and designed the study protocol. FBW planned the statistical analysis and data extraction. RH and VC provided critical insights. All authors have approved and contributed to the final written manuscript.

**Funding** This research received no specific grant from any funding agency in the public, commercial or not-for-profit sectors.

**Competing interests** None declared.

**Patient consent** Not required.

**Provenance and peer review** Not commissioned; externally peer reviewed.

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
