## [Reviewer comments · BMJ Open]

ARTICLE DETAILS

TITLE (PROVISIONAL)	Determinants of non-adherence to treatment for tuberculosis in high- and middle-income settings: a systematic review protocol
AUTHORS	Wurie, Fatima; Cooper, Vanessa; Horne, Robert; Hayward, Andrew

VERSION 1 – REVIEW

REVIEWER	Dr David J Roberts MBChB MSc MRCP MFPH Public Health Specialty Registrar, Oxford School of Public Health, New Richards Building, Old Road Campus, Headington, Oxford, OX3 7LG
REVIEW RETURNED	03-Oct-2017

GENERAL COMMENTS	NEED FOR THIS STUDY: The study could make a worthwhile addition to the literature. ABSTRACT - This does not mention the authors seek to perform a synthesis of both quantitative and qualitative studies. This is important because the two types of studies demand rather different methods for synthesis. It also states GRADE will be used to assess study quality, a use for which GRADE is not designed (see comments below). INTRO - the sentence 'these findings suggest that the effectiveness of TB treatment is considerably poor' does not make sense from a grammatical or clinical point of view. The 'Necessity-Concerns Framework' is introduced but then there is no further explanation of what it is or its relevance to the study. Consider more international references that reflect the wider BMJ Open readership outside of the UK. I'm not sure the measures to support adherence paragraph is all that useful in framing the rationale for the study, unless the various measures are explicitly associated with the relevant constructs of adherence/adherence behaviour by which they might work. STUDY OBJECTIVES - This does not adequately reflect the mixed quantitative and qualitative nature of the study (it currently seems very quantitative focused). A qualitative synthesis will seek to answer a different focus from the current question e.g. what factors did patients/clinicians feel were important determinants of non-adherence; this is not the same as asking which factors are quantitatively associated with non-adherence.
---

STUDY DESIGN: This study design is most appropriate to answer a quantitative research question. Despite stating qualitative studies will be included, there is no mention of the specific methods usually employed for their quality assessment or synthesis. This must be addressed.

SEARCH STRATEGY: Is an expert librarian to be consulted on the search strategy? What about grey literature? A draft strategy for one database (typically MEDLINE) should be included in the protocol.

SELECTION CRITERIA: the authors state they will include '...predictors found to have increased uptake, adherence and continuation of treatment'. These are three different aspects/constructs of adherence. The authors should more clearly define what they understand by 'adherence' in terms of these constructs, in order to explain their study selection criteria (specifically, the outcomes of interest in the target literature). It may also provide another framework for synthesis by the constructs of interest. Will limits in terms of language or year of publication be applied?

SELECTION PROCESS: I understand 2 authors will independently screen titles/abstracts, but it is not clear how they will resolve any conflicts at this stage (it is stated for the next stage, but I shouldn't need to assume the same thing was done at the first stage).

DATA EXTRACTION:The authors state they will use a 'thematic analysis to synthesise findings from qualitative studies'. This is a synthesis method, not a data extraction method. More detail is needed to understand the process planned to extract data from qualitative studies e.g. are first order and second order constructs both to be extracted? For both quantitative and qualitative studies, Is there a primary outcome?

RISK OF BIAS ASSESSMENT: GRADE is not a method for assessing risk of bias for a single study, but instead a method to describe and summarize the certainty of evidence for a given outcome, across studies. The authors need to specify use of a risk of bias tool that will be used to judge each study individually, and is appropriate for the study designs to be included, including observational and qualitative studies. They should also state whether two authors will do this independently (the norm). For quantitative studies, explicit consideration of the method used to measure adherence (and its validity) need to be considered, particularly self-reported measures versus more robust/objective methods such as (electronic) pill counts.

REPORTING BIAS: how will the authors assess for the risk of reporting bias?

DATA SYNTHESIS: Again, specific methods for synthesis of qualitative data are not described. For quantitative studies, how will the authors deal with different study designs, different lengths of follow up, and different effect measures or different instruments (e.g. self report and pharmacist pill count) used for the same construct? What measures of effect of the factor of interest will be used e.g. risk ratio, mean difference, hazard ratio? How will risk of bias findings be summarized?

	META-ANALYSIS: What I2 will be required before random effects modelling is used over fixed effects? Will causes of heterogeneity be explored by further subgroup analyses? What software will be used? OTHER OUTSTANDING ISSUES: Will the GRADE approach be used after synthesis to summarize the certainty of evidence for the outcomes of interest? How will they deal with observational studies, for which GRADE is not specifically designed? Will the authors perform a sensitivity analysis, taking account of individual study quality? PRISMA: The authors state the protocol was prepared to PRISMA guidelines, but not that it will also be adhered to when reporting the final study. I note the protocol is registered on PROSPERO.
--	--

REVIEWER	Dennis Falzon WHO Global TB Programme, Avenue Appia 20, CH1211 Geneva Switzerland I am involved in the development of policy recommendations on TB treatment adherence, including digital health products
REVIEW RETURNED	13-Oct-2017

GENERAL COMMENTS	p7, Search strategy: it would be helpful if you cite the exact terms to be used here You will not be querying individual patient datasets (e.g. ECDC's TESSy database), which could be a valuable resource for such studies and allow adjustment for several co-variables. If this is not envisaged it needs to be listed as a limitation and maybe a source for future studies /research. p7, line 1: what does "immune status" mean? TST/IGRA? BCG vaccination? or past treatment history? p7, line 2: I trust you will stratify by the drug-resistant strain status (e.g. MDR/non-MDR) of the study population p7, line 3: can you specify that the treatment regimen and maybe individual drugs (e.g. injectables) will be assessed as one of the determinants of adherence?
--

VERSION 1 – AUTHOR RESPONSE

Reviewer: 1

Reviewer Name: Dr David J Roberts MBChB MSc MRCP MFPH

Institution and Country: Public Health Specialty Registrar, Oxford School of Public Health, New Richards Building, Old Road Campus, Headington, Oxford, OX3 7LG

Please state any competing interests: None declared

Please leave your comments for the authors below

NEED FOR THIS STUDY: The study could make a worthwhile addition to the literature.

ABSTRACT - This does not mention the authors seek to perform a synthesis of both quantitative and qualitative studies. This is important because the two types of studies demand rather different methods for synthesis. It also states GRADE will be used to assess study quality, a use for which GRADE is not designed (see comments below).

Response: We have now decided to perform only a synthesis of quantitative studies. We accept that the standard

GRADE is not the most appropriate tool to assess strength of evidence across studies of risk factors as it considers observational study designs as being weaker than trials. Given trials are not suitable for evaluation of risk factors, this would not be appropriate. For cross-study assessment of strength of evidence for particular risk factors affecting adherence we will use the GRADE tool modified for Prognostic studies. <http://www.bmj.com/content/350/bmj.h870.full.printacross> studies This considers observational studies as the most appropriate for assessing the frequency of, and risk factors for, particular outcomes (in this case adherence outcomes).

INTRO - the sentence 'these findings suggest that the effectiveness of TB treatment is considerably poor' does not make sense from a grammatical or clinical point of view. The 'Necessity-Concerns Framework' is introduced but then there is no further explanation of what it is or its relevance to the study. Consider more international references that reflect the wider BMJ Open readership outside of the UK. I'm not sure the measures to support adherence paragraph is all that useful in framing the rationale for the study, unless the various measures are explicitly associated with the relevant constructs of adherence/adherence behaviour by which they might work.

Response: The sentence has been adjusted. Given that we will not be conducting a qualitative synthesis we have removed the sentence about the NCF. Whilst we have removed the paragraph about the various approaches to measure adherence from the introduction section, we have included these measures to help determine our outcome on page 6 of the manuscript.

STUDY OBJECTIVES - This does not adequately reflect the mixed quantitative and qualitative nature of the study (it currently seems very quantitative focused). A qualitative synthesis will seek to answer a different focus from the current question e.g. what factors did patients/clinicians feel were important determinants of non-adherence; this is not the same as asking which factors are quantitatively associated with non-adherence.

Response: We have removed the qualitative study review.

STUDY DESIGN: This study design is most appropriate to answer a quantitative research question. Despite stating qualitative studies will be included, there is no mention of the specific methods usually employed for their quality assessment or synthesis. This must be addressed.

Response: We have removed the qualitative study review.

SEARCH STRATEGY: Is an expert librarian to be consulted on the search strategy? What about grey literature? A draft strategy for one database (typically MEDLINE) should be included in the protocol.

Response: We plan to seek expertise and advice from an expert librarian. We have included a search strategy for Medline as a supplementary file. We do not plan to use grey literature.

In **SELECTION CRITERIA**, the authors state they will include '...predictors found to have increased uptake, adherence and continuation of treatment'. These are three different aspects/constructs of adherence. The authors should more clearly define what they understand by 'adherence' in terms of these constructs, in order to explain their study selection criteria (specifically, the outcomes of interest in the target literature). It may also provide another framework for synthesis by the constructs of interest. Will limits in terms of language or year of publication be applied?

Response: We will focus our review on measures of adherence as our primary outcome measure. A complication is the wide range of measures of adherence and the fact that measuring adherence may improve adherence. From page 5 to 6 of the manuscript we have stated that "...non-adherence will be determined by self-reporting through attendance at follow-up appointments, collecting prescriptions from clinics, pill counts and pharmacy reports, electronic devices (medication events monitoring systems (MEMS) caps), urine inspection, testing for drug levels and directly-observed therapy (DOT) attendance or video-observed therapy (VOT) sessions. Studies that report outcomes such as non-completion of treatment and/or lost to follow-up and/or treatment refusal will also be included."

SELECTION PROCESS: I understand 2 authors will independently screen titles/abstracts, but it is not clear how they will resolve any conflicts at this stage (it is stated for the next stage, but I shouldn't need to assume the same thing was done at the first stage).

Response: We have amended this paragraph to reflect that both authors will be involved in independently screening titles and abstracts as well as full-text articles.

DATA EXTRACTION:The authors state they will use a 'thematic analysis to synthesise findings from qualitative studies'. This is a synthesis method, not a data extraction method. More detail is needed to understand the process planned to extract data from qualitative studies e.g. are first order and second order constructs both to be extracted? For both quantitative and qualitative studies, Is there a primary outcome?

Response: This review will now only focus on the synthesis of quantitative studies.

RISK OF BIAS ASSESSMENT: GRADE is not a method for assessing risk of bias for a single study, but instead a method to describe and summarize the certainty of evidence for a given outcome, across studies. The authors need to specify use of a risk of bias tool that will be used to judge each study individually, and is appropriate for the study designs to be included, including observational and qualitative studies. They should also state whether two authors will do this independently (the norm). For quantitative studies, explicit consideration of the method used to measure adherence (and its validity) need to be considered, particularly self-reported measures versus more robust/objective methods such as (electronic) pill counts.

Response: We will use risk of bias domains used for the Cochrane Collaboration for specific observational study designs. Assessment of risk of bias of individual studies and outcomes will be conducted by two reviewers independently and will subsequently discuss amongst all authors for arbitration. For cross-study assessment of strength of evidence for particular risk factors affecting adherence we will use the GRADE tool modified for Prognostic studies.

<http://www.bmj.com/content/350/bmj.h870.full.printacross studies>

REPORTING BIAS: how will the authors assess for the risk of reporting bias?

Response: We will prioritise objective measures of adherence that are less prone to reporting bias

DATA SYNTHESIS: Again, specific methods for synthesis of qualitative data are not described. For quantitative studies, how will the authors deal with different study designs, different lengths of follow up, and different effect measures or different instruments (e.g. self report and pharmacist pill count) used for the same construct? What measures of effect of the factor of interest will be used e.g. risk ratio, mean difference, hazard ratio? How will risk of bias findings be summarized?

Response: We will now not include a qualitative synthesis.

We anticipate a high proportion of studies will either report adherence during the first two months (initiation phase of treatment) or throughout treatment and that a wide variety of instruments will be used. We will report 2 and 6 month adherence measures separately. We will report different instruments separately. We will also highlight whether there is consistency of identified risk factors across instruments and for the different time periods.

META-ANALYSIS: What I² will be required before random effects modelling is used over fixed effects? Will causes of heterogeneity be explored by further subgroup analyses? What software will be used?

Response: A p value for I² of less than 0.05 will indicate that heterogeneity among the group of studies being analysed was significant. If the I² statistic is greater than 50% (with p<0.05) for each treatment outcome, a random-effects analysis, incorporating the impact of both chance and heterogeneity among study populations and study design, will be chosen over the fixed-effects alternative, which assumes that differences among study outcomes are due entirely to chance. We will use STATA to conduct our meta-analysis.

OTHER OUTSTANDING ISSUES: Will the GRADE approach be used after synthesis to summarize the certainty of evidence for the outcomes of interest? How will they deal with observational studies, for which GRADE is not specifically designed? Will the authors perform a sensitivity analysis, taking account of individual study quality?

Response: See above for use of GRADE approach modified for prognostic studies.

PRISMA: The authors state the protocol was prepared to PRISMA guidelines, but not that it will also be adhered to when reporting the final study. A completed PRISMA checklist is included in this submission. We will adhere to PRISMA guidelines in the reporting of the final study results.

I note the protocol is registered on PROSPERO.

Response: We will update PROSPERO with the revised methodology

Reviewer: 2

Reviewer Name: Dennis Falzon

Institution and Country: WHO Global TB Programme, Avenue Appia 20, CH1211 Geneva, Switzerland

Please state any competing interests: None declared

I am involved in the development of policy recommendations on TB treatment adherence, including digital health products

Please leave your comments for the authors below

p7, Search strategy: it would be helpful if you cite the exact terms to be used here

Response: We have included a copy of the search strategy used for Medline as a supplementary file as part of this submission.

You will not be querying individual patient datasets (e.g. ECDC's TESSy database), which could be a valuable resource for such studies and allow adjustment for several co-variables. If this is not envisaged it needs to be listed as a limitation and maybe a source for future studies /research. As a systematic review of existing published evidence we will not be including de-novo analysis of individual patient datasets.

p7, line 1: what does "immune status" mean? TST/IGRA? BCG vaccination? or past treatment history?

Response: We have amended the characteristics of the study population.

p7, line 2: I trust you will stratify by the drug-resistant strain status (e.g. MDR/non-MDR) of the study population.

Response: We will conduct separate analyses of MDRTB patients as well as considering MDRTB as a risk factor for adherence outcomes.

p7, line 3: can you specify that the treatment regimen and maybe individual drugs (e.g. injectables) will be assessed as one of the determinants of adherence?

Response: We will perform sub-analyses to assess whether treatment regimens are predictors of non-adherence.